# Active Learning with the nnUNet and Sample Selection with Uncertainty-Aware Submodular Mutual Information Measure

**Bernhard Föllmer**[1]                                      BERNHARD.FOELLMER@CHARITE.DE
**Kenrick Schulze**[1]                                       KENRICK.SCHULZE@CHARITE.DE
**Christian Wald**[2]                                        WALD@MATH.TU-BERLIN.DE
**Sebastian Stober**[3]                                      STOBER@OVGU.DE
**Wojciech Samek**[4,5,6]                        WOJCIECH.SAMEK@HHI.FRAUNHOFER.DE
**Marc Dewey**[1,7]                                          MARC.DEWEY@CHARITE.DE

[1] *Department of Radiology, Charité-Universitätsmedizin Berlin, corporate member of Freie Universität Berlin and Humboldt-Universität zu Berlin, Berlin, 10117, Germany*

[2] *Institute of Mathematics, Technical University of Berlin, Berlin, Germany*

[3] *Artificial Intelligence Lab, Otto-von-Guericke-Universität, Magdeburg, Germany*

[4] *Department of Artificial Intelligence, Fraunhofer Heinrich Hertz Institute, Berlin, Germanyy*

[5] *BIFOLD – Berlin Institute for the Foundations of Learning and Data, Berlin, Germany*

[6] *Department of Electrical Engineering and Computer Science, Technical University of Berlin, Berlin, Germanyy*

[7] *Berlin Institute of Health and DZHK (German Centre for Cardiovascular Research), partner site Berlin, Germany and Deutsches Herzzentrum der Charité (DHZC), Berlin, Germany*

**Editors:** Accepted for publication at MIDL 2024

## Abstract

Annotating medical images for segmentation tasks is a time-consuming process that requires expert knowledge. Active learning can reduce this annotation cost and achieve optimal model performance by selecting only the most informative samples for annotation. However, the effectiveness of active learning sample selection strategies depends on the model architecture and training procedure used. The nnUNet has achieved impressive results in various automated medical image segmentation tasks due to its self-configuring pipeline for automated model design and training. This raises the question of whether the nnUNet is applicable in an active learning setting to avoid cumbersome manual configuration of the training process and improve accessibility for non-experts in deep learning-based segmentation. This paper compares various sample selection strategies in an active learning setting in which the self-configuring nnUNet is used as the segmentation model. Additionally, we propose a new sample selection strategy for UNet-like architectures: **USIM - U**ncertainty-Aware **S**ubmodular Mutual **I**nformation **M**easure. The method combines uncertainty and submodular mutual information to select batches of uncertain, diverse, and representative samples. We evaluate the performance gain and labeling costs on three medical image segmentation tasks with different segmentation challenges.. Our findings demonstrate that utilizing nnUNet as the segmentation model in an active learning setting is feasible, and most sampling strategies outperform random sampling. Furthermore, we demonstrate that our proposed method yields a significant improvement compared to existing baseline methods.

**Keywords:** Deep Learning, Active Learning, Medical Image Segmentation, nnUNet, Submodular Subset Selection

## 1. Introduction

Segmentation of tumors and surrounding anatomy is an important task in medical imaging for cancer diagnosis using CT and MRI (Hesamian et al., 2019). However, training these models requires a large number of voxel-wise labeled images. Annotating such datasets is usually time-consuming and requires expert knowledge (Ren et al., 2022).

Active learning (AL) can address this issue by iteratively selecting only the most informative samples (batches) from the unlabeled dataset for annotation and training, while still achieving near-optimal model performance. However, the performance of AL sampling strategies depend heavily on the model architecture and training procedure used (Ji et al., 2023). To analyze the gain of AL strategies, it is important to ensure robustness and reproducability under different experimental conditions (Munjal et al., 2022). The performance gain and labeling cost reduction of AL strategies depend on the following factors. First, the training pipeline including data preprocessing (normalization, augmentation), model architecture (patch size, number of layers), and training procedure (batch size, learning rate scheduling). Second, the dataset characteristics such as dataset size, class imbalance, difficulty of the learning task. Third, the labeling budget during each sampling round.

The nnUNet (Isensee et al., 2021) is a self-configuring pipeline, which allows for the training of 2D and 3D UNet-like models (Ronneberger et al., 2015) without the need for manual adaptation of preprocessing, model architecture, or hyperparameters and has won many segmentation competitions (Antonelli et al., 2022). The nnUNet identifies robust design decisions based on multiple tasks (Isensee et al., 2021), ensuring reproducibility and robustness in evaluating AL strategies (Munjal et al., 2022; Burmeister et al., 2022) by automatically configuring preprocessing, model architecture, and hyperparameters. This paper investigate whether the use of a self-configuring training pipeline in an AL setting can reduce annotation costs. This would also increase reproducibility and could facilitate the use of active learning for medical image segmentation.

Most deep AL strategies that combine uncertainty, diversity and representativeness in their sampling objective have been developed for classification tasks, and less frequently for segmentation tasks (Saidu and Csató, 2021). Novel AL strategies such as BADGE (Ash et al., 2020) select samples based on diverse gradients where gradient length captures the uncertainty. We explore the adaptation of gradient-based active learning methods to U-Net-like architectures and segmentation tasks with high-dimensional annotations. To efficiently combine predictive uncertainty and gradient based sample representation, we propose **USIM - U**ncertainty-aware **S**ubmodular mutual **I**nformation **M**easure. This approach combines predictive uncertainty-based sampling with diversity and representative sampling in parameter space using submodular mutual information measures (Kothawade et al., 2021, 2022a). The novelty of our approach lies in the selection of the query set based on class-weighted predictive uncertainty using Monte Carlo Dropout and the estimation of representative gradient embeddings based on the bottleneck layer (USIMC) as well as an automated gradient embedding selection based on the Fisher information (USIMF).

Our contributions are twofold: First, we evaluate various active learning strategies using the nnUNet pipeline as the segmentation model in an AL setting. Second, we propose and evaluate USIM, an AL strategy that combines predictive uncertainty with diversity and representativeness using a submodular mutual information measure.

## 2. Related work

Two main principles for selecting batches of informative samples are uncertainty-based sampling and representation-based sampling. Various uncertainty-base approaches exist (Pratapa et al., 2011; Li and Alstrøm, 2020), such as Monte Carlo dropout-based methods (Gal and Uk, 2016; Kendall et al., 2017), Bayesian neural networks (Gal et al., 2017), or ensemble methods (Chitta et al., 2018). However, methods that rely solely on uncertainty are not suitable for large datasets with redundant information.

Representation-based sampling methods model the representativeness and diversity of samples within a batch. For example, the Core-set approach (Sener and Savarese, 2018) estimates distances between samples modeled by the Euclidean distance between feature vectors. Uncertainty and representation-based methods have been used for classification and segmentation tasks (Burmeister et al., 2022).

Hybrid methods aim to combine uncertainty and diversity in their sampling objectives to select informative samples while avoiding redundancy (Yang et al., 2017; Nath et al., 2021). These methods rely on compact image representation which are typically extracted from the last layer of networks for classification tasks such as BADGE (Ash et al., 2020). Sreenivasaiah et al. investigated adaptation of this method for segmentation tasks (Aklilu and Yeung, 2022). MEAL (Sreenivasaiah et al.) extracts embeddings based on Uniform Manifold Approximation to model representativeness of image patches for image segmentation.

In addition, semi-supervised approaches offer additional solutions to reduce labeling costs (Mittal et al., 2023; Gaillochet et al., 2022). Recent active learning strategies make use of the Fisher information ratio, Hessian, or similarity matrices but were mainly developed for classfication task (Kirsch and Gal, 2022; Kothawade et al., 2021; Ash et al., 2020, 2021; Liu et al., 2021) and seldom for segmentation tasks (Al, 2019; Yu et al., 2023).

We explicitly compare practical and user-friendly methods and exclude methods that require design of additional models to evaluate informativeness of unlabeled samples, such as VAAL (Sinha et al., 2019) or adaptation of the loss function. We further exclude methods that require additional sub-networks with trainable parameters that might influence the automated architecture configuration of the nnUNet due to memory restrictions (Yoo and Kweon, 2019).

## 3. Method

### 3.1. Active learning and submodular subset selection

Active learning is the process where the learning algorithm attempts to maximize a model's performance gain while annotating the fewest samples possible (Ren et al., 2022).

#### 3.1.1. ACTIVE LEARNING FOR MEDICAL IMAGE SEGMENTATION

To evaluate AL sampling strategies for medical image segmentation, the dataset is split into training and test set. From the training data, we construct an unlabeled pool $X_U = \{x^{(j)}\}_{j=1}^{U}$ of patches $x \in \mathbb{R}^{C \times H \times W}$ with patch width $W$, height $H$ and number of channels $C$. We randomly select and label a small initial labeled dataset $X_L = \{(x^{(j)}, y^{(j)})\}_{j=1}^{L}$ with segmentation masks $y \in \mathbb{R}^{K \times H \times W}$ ($K$ - number of classes) to train the segmentation model. During multiple sampling rounds, we use one of the sampling strategies to select a batch

$X_B \subseteq X_U$ of $B$ most informative samples, simulate annotation and add the annotated samples to the labeled dataset. Since retraining from scratch is very time consuming, we fine-tune the model in each AL round and evaluate the model using the test set.

### 3.1.2. Submodular mutual information measure for subset selection

Submodular functions are a class of functions that can be used to model guided (uncertainty-guided in our approach) data subset selection of representative subsets (Kothawade et al., 2022a). The submodular mutual information (SMI) function is given by $I_f(X_B; X_Q) = f(X_B) + f(X_Q) - f(X_B \cup X_Q)$, where $X_Q$ is the query set (target set), $X_B$ is the selected subset of patches from the unlabeled data set and $f : 2^X \to \mathbb{R}$ a submodular set function defined as $f(X) = \sum_{x_i \in \Omega} \max_{x_j \in X} S_{ij}$. The term $S_{ij}$ measures the similarity between the elements $x_i$ and $x_j$. The instantiated submodular function $I_f(X_B; X_Q)$ is used to select a subset that maximizes the submodular function $X_B \leftarrow \text{argmax}_{X_B \subseteq X_U, |X_B| \leq B} I_f(X_B; X_Q)$. Intuitively, SMI models the similarity between $X_Q$ and $X_B$, and maximizing SMI will select points similar to $X_Q$ while being diverse. We use the mutual information function variant for the facility location (FL) function defined over $X_Q$ (FLQMI) shown in Equation 1. The variant was successfully applied in Kothawade et al. (2022a) and is memory and time efficient. The parameter $\eta$ balances query-relevance and diversity. Kothawade et al. (2022b) and Beck et al. (2024) performed a qualitative an theoretical analysis with respect to query-coverage, query-relevance and diversity, and showed that as soon as $\eta$ is increased, the summary produced by FLQMI becomes more query-relevant and less diverse.

$$I_f(X_B; X_Q) = \sum_{x_i \in X_Q} \max_{x_j \in X_B} S_{ij} + \eta \sum_{x_i \in X_B} \max_{x_j \in X_Q} S_{ij} \tag{1}$$

We further investigate the robustness of our method in a redundancy scenario to confirm the superiority of the submodular information measure for subset selection compared to sampling with a uniform distribution over the query set in Appendix E.

### 3.2. Active learning with the self-configuring nnUNet pipeline

We first run the nnUnet preprocessing and planning function on the raw dataset to generate a dataset fingerprint and automatically configure model architecture and training procedure. Based on the automatic configured parameters (patch size, resampling parameter, etc.) we extract overlapping image patches to create the unlabeled pool of samples. We label a small randomly selected subset and run the self-configuring nnUNet pipeline again to re-configure a subset of parameters for resampling, intensity normalization and train the initial model. We used the trained model to select the most informative batch of samples using one of the compared sample selection strategies. After annotation, we rerun the nnUNet pipeline again to update the dataset fingerprint and planing configuration, and finetune the model. Since the characteristics of the image data remain the same and only the annotations change, most model configurations, such as model architecture and learning rate, remain unchanged. The only changes that might occur are annotation resampling and intensity normalization. Sampling, annotation and self-configured model finetuning is repeated in multiple AL round. The proposed AL framework is shown in Figure 1.

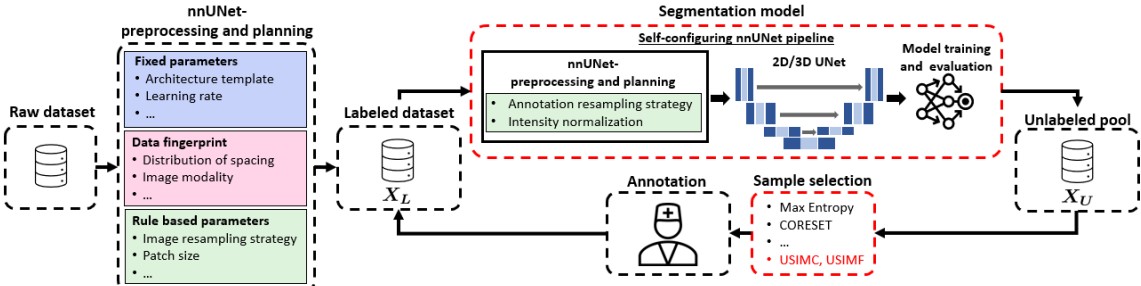

Figure 1: Active learning utilizing the self-configuring nnUNet pipeline. The nnUNet-pipeline automatically configures and trains the model in each active learning round. The most informative samples are selected from the unlabeled dataset using one of the compared sampling strategies including our proposed USIMC and USIMF methods. Images are annotated by the radiologist (oracle) and added to the labeled dataset. The procedure is repeated in multiple AL rounds to increase model performance while annotating the fewest samples possible.

### 3.3. USIM - Uncertainty-aware submodular mutual information measure for subset selection

We propose USIM, a sample selection strategy that combines predictive uncertainty and submodular mutual information measure for subset selection with UNet-like architectures.

#### 3.3.1. PREDICTIVE UNCERTAINTY AS A MEASURE OF INFORMATIVENESS

To measure the informativeness of a sample, predictive uncertainty using Monte Carlo dropout is a powerful and simple method (Li and Alstrøm, 2020). In our work, the uncertainty of the unlabeled samples is estimated using Monte Carlo dropout and the model $\boldsymbol{\theta}_{k-1}$ trained in the previous round. A subset of $Q$ uncertain samples $\boldsymbol{X_Q}$ is sampled (with replacement) from the unlabeled dataset with sampling probability proportional to a weighted uncertainty $p(\boldsymbol{\theta}_{k-1}, \boldsymbol{x}) = \frac{\sum_c w_c \cdot Uncert(\boldsymbol{\theta}_{k-1}, \boldsymbol{x}, c)}{\sum_{\boldsymbol{x} \in \boldsymbol{X_U}} \sum_c w_c \cdot Uncert(\boldsymbol{\theta}_{k-1}, \boldsymbol{x}, c)}$. The uncertainty of a foreground class $c$ is weighted by $w_c$ which is the inverse of the number of annotated voxels of that class in the training set. The uncertainty-based weighting ensures that the performance is less biased towards classes where more data is presented. We analyze the influence of query set size $Q$ in Appendix D. In our experiments, $Q$ is estimated using the elbow method (Thorndike, 1953) which plots the sample uncertainties in descending order and takes the elbow of the curve as the number of uncertain samples in the data set.

#### 3.3.2. MUTUAL INFORMATION BASED SUBMODULAR SUBSET SELECTION WITH GRADIENT EMBEDDINGS

To create an informative batch of samples, it is important to consider not only uncertainty but also diversity and representativeness. Since we select the query set $\boldsymbol{X_Q}$ by sampling from the unlabeled dataset with a probability distribution proportional to the class-weighted uncertainty, the query set includes not only most uncertain but more diverse samples. To

confirm that the proposed sampling strategy of the query set is more efficient than standard active learning (query set is the unlabeled dataset) (Kothawade et al., 2021) or sampling the most uncertain samples from the unlabeled dataset, we compare the performance gain for the methods in Appendix D. In our approach, we use mutual information function variant for the facility location (FL) function (FL**Q**MI) show in Equation 1. The method requires a similarity matrix $\boldsymbol{S_{uq}} \in \mathbb{R}^{|\boldsymbol{X_U}| \times |\boldsymbol{X_Q}|}$ between samples of the unlabeled dataset and the query set and can be constructed in many ways, e.g. by computing the cosine distance (Kothawade et al., 2022a), Euclidean distance (Neven and Goedemé, 2023) or Fisher information kernel. We analyze two variants of similarity matrices using cosine similarity between gradient embeddings of the bottleneck layer defined as **USIMC** and an approximation of pairwise influence between samples using the Fisher kernel which is defined as **USIMF**. Details about the construction of the similarity matrices can be found in Appendix A. After construction of the similarity matrix, we instantiate the submodular function $I_f(\boldsymbol{X_B}; \boldsymbol{X_Q})$ and use stochastic greedy method to select the batch $\boldsymbol{X_B}$ for labeling. We use the stochastic greedy optimizer from SUBMODLIB (Kaushal et al., 2022) because it has a provably linear running time independent of the budget and is faster than the naive greedy approach. We summarize the proposed USIM method in Algorithm 1.

---

**Algorithm 1:** USIM - Uncertainty-Aware Submodular Mutual Information Measure

1: **Input:** Initial labeled dataset: $\boldsymbol{X_L}$, unlabeled dataset: $\boldsymbol{X_U}$, initial self-configured nnUNet model parameter: $\boldsymbol{\theta}_0$, batch size: $B$, number of sampling rounds: $K$

2: **for** $k = 1, 2, ..., K$ **do**

3:     Estimate sample uncertainty $Uncert(\boldsymbol{\theta}_{k-1}, \boldsymbol{x}) \in \mathbb{R}$ for all unlabeled samples $x \in \boldsymbol{X_U}$

4:     Sample a query subset $\boldsymbol{X_Q} \subseteq \boldsymbol{X_U}$ with sample probability proportional to weighted uncertainty

5:     Compute similarity matrix $\boldsymbol{S_{uq}}$ with Equation 2 for USIMC or 4 for USIMF

6:     Instantiate the submodular function $I_f(\boldsymbol{X_B}; \boldsymbol{X_Q})$ based on $\boldsymbol{S_{uq}}$ (Equation 1)

7:     Use stochastic greedy method to select batch $\boldsymbol{X_B}$ with
    $\boldsymbol{X_B} \leftarrow \mathrm{argmax}_{\boldsymbol{X_B} \subseteq \boldsymbol{X_U}, |\boldsymbol{X_B}| \leq B} I_f(\boldsymbol{X_B}; \boldsymbol{X_Q})$

8:     Query the oracle to obtain segmentation masks $\boldsymbol{y}(\boldsymbol{x}), \forall \boldsymbol{x} \in \boldsymbol{X_B}$

9:     $\boldsymbol{X_L} \leftarrow \boldsymbol{X_L} \cup \boldsymbol{X_B}$ ; $\boldsymbol{X_U} \leftarrow \boldsymbol{X_U} \setminus \boldsymbol{X_B}$

10:    Train model on $\boldsymbol{X_L}$: $\boldsymbol{\theta}_k = \arg \min_{\boldsymbol{\theta}} \mathbb{E}_{\boldsymbol{X_L}}[l(\boldsymbol{x}, \boldsymbol{y}; \boldsymbol{\theta})]$

11: **end for**

12: **return** Final model $\boldsymbol{\theta}_K$

---

## 4. Experiments and Results

### 4.1. Datasets

We evaluate the AL sampling methods on three medical image segmentation datasets from the Medical Segmentation Decathlon (Antonelli et al., 2022) to ensure method evaluation on different target regions, modalities and challenging features. For our analysis, we choose the labeled training set of 1) Spleen dataset, as a small dataset of 41 CT scans with a large ranging foreground size, 2) Liver dataset, a large dataset of 131 CT scans with label imbalance between large liver class and small tumor class and 3) Hippocampus dataset

consisting of 260 MRI scans with the challenge to segment two adjacent small structures with high precision

## 4.2. Active learning sampling strategies and implementation

**Sampling Strategies:** We compare the following sampling method in our analysis: (1) Random Sampling, (2) Max Entropy (Shannon, 1948), (3) Mean STD (Kendall et al., 2017), (4) BALD (Gal et al., 2017), (5) Core-Set (Sener and Savarese, 2018), (6) BADGE(LL) (Ash et al., 2020; Aklilu and Yeung, 2022), (7) Stochastic Batches (Gaillochet et al., 2023) and (8) USIMC (ours), (9) USIMF (ours).

**Implementation:** For our experiments, we used 2D nnUNet configurations because they have a higher training speed and are less prone to overfitting. However, the proposed USIM method can easily be extended to 3D models and will be studied in further research.

For all our experiments, we set $\eta = 1.0$ and analyzed the influence of the hyperparameter in Appendix D. The used hardware configurations and self-configured hyperparameters can be found in Table 1 in Appendix F. The code, for training and evaluation is available at https://github.com/Berni1557/ALUNET.

Further details about sampling strategies and implementation can be found in Appendix C.

## 4.3. Results

The performance gain in terms of dice score with respect to the number of annotated samples for all competing strategies is shown in Figure 2.

**Spleen:** All sampling strategies except Core-Set outperformed random sampling. However, the segmentation task is less challenging and the performance differences between the strategies are therefore small. With less than 8% of annotated data, USIMF achieved a similar dice score as trained with fully annotated dataset.

**Liver:** Uncertainty-based methods (BALD, Mean STD) performed better than random sampling. USIMF and USIMC outperformed most methods and reach near optimal performance with less than 5% of the data.

**Hippocampus:** USIMC and USIMF outperformed all other sampling strategies and reach performance of the fully annotated dataset with roughly 30% of the annotated data. All other strategies outperformed or performed similar to random sampling.

The pairwise penalty matrix (Ji et al., 2023) in Figure 2 (D) aggregates results over all conducted experiments. The overall performance $\Phi$ is measured by the column-wise average, where lower numbers indicate a higher-performing algorithm. The results show that USIMC and USIMF have the lowest column sum, indicating that they outperform other strategies. The Figures 10, 11, and 12 in Appendix G show examples of AL based segmentation results for all three datasets. Figure 3 shows a t-SNE plot of the USIMF gradient embeddings on the Liver dataset.

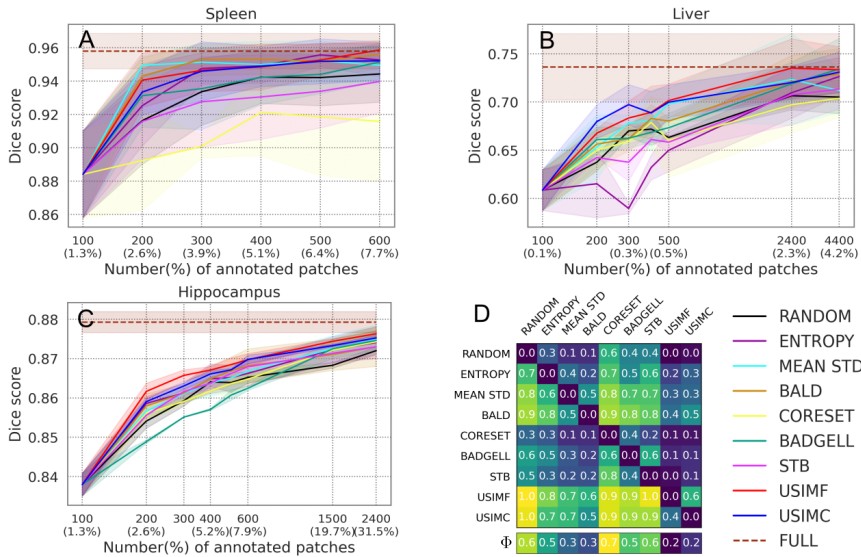

Figure 2: Performance comparison of active learning strategies (Random sampling, Maximum Entropy, Mean Std, BALD, Core-Set, BADGE(LL), Stochastic Batches, USIMC (ours), USIMF (ours) and fully labeled dataset for the Spleen (A), Liver (B) and Hippocampus (C) dataset. A pairwise penalty matrix is shown in D. Element i j corresponds to the number of times (expressed in percentage) algorithm i outperforms algorithm j. Column-wise averages Φ are given where a lower number corresponds to a higher-performing algorithm.

## 5. Discussion and Conclusion

In this paper, we have investigated the utility of the nnUNet as a self-configuring model in an AL setting to reduce labeling costs. We additionally proposed USIM, an AL strategy that combined predictive uncertainty and submodular mutual information measure to select informative, diverse, and representational batches with two similarity matrix variants (USIMF and USIMC). The experiments confirmed that using the self-configuring nnUNet pipeline in an active learning setting is an effective strategy for reducing labeling costs and facilitating AL by avoiding the cumbersome configuration of the training process. All methods performed equally or better than random sampling. We showed that methods based on uncertainty (BALD, STD MEAN) outperformed those based on representation (Core-Set). Our hybrid method outperformed the compared active learning methods, with USIMF performing slightly better than USIMC. The proposed method was evaluated only on 2D nnUNets, but it can be extended to 3D approaches. However, we refrained from conducting the evaluation due to longer training times, leaving it for future research. Further research is necessary to prove the effectiveness of active learning for medical image segmentation in real scenarios, rather than relying solely on simulations.

## Acknowledgments

This work was funded by the German Research Foundation through the graduate program BIOQIC (GRK2260, project-ID: 289347353).

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

## Appendix A. Similarity matrices for submodular mutual information measure

We analyze two gradient based similarity matrices for the submodular mutual information measure.

**USIMC**: In USIMC, we used the cosine similarity between gradient embeddings of bottleneck layer as similarity measure to construct a similarity matrix $S_{uq}$. The matrix measures similarity between the gradients of the unlabeled dataset $g_u$ and the query set $g_q$.

$$S_{uq} = \frac{g_u \cdot g_q}{\|g_u\| \|g_q\|} \tag{2}$$

**USIMF**: In USIMF, we use the approximated pairwise influence (Fisher kernel) as similarity matrix. The Fisher kernel, has already been used in Liu et al. (2021) for influence selection-based active learning or for deep active learning on biased datasets (Gudovskiy et al., 2020). The similarity matrix $S_{uq} \in \mathbb{R}^{|X_U| \times |X_Q|}$ is constructed between the unlabeled dataset $X_U$ and the query data set $X_Q$. To select only the gradients from important parameters, we use a simple method that selects parameters based on the empirical Fisher information matrix (Tu et al., 2016). The use of the Fisher matrix to identify model parameters that are important for a learning task was proposed by Kirkpatrick et al. (2017) to avoid catastrophic forgetting. We use Equation 3 to approximate the Fisher information matrix based on randomly sampled subset of the query set $X_Q$ with pseudo labels $\bar{y}$ and predicted class probability $p_{\theta}(\bar{y}|x)$.

$$\mathcal{F}(\boldsymbol{\theta}) = \text{diag}(\frac{1}{|X_Q|} \sum_{x \in X_Q} \nabla_{\boldsymbol{\theta}} \log p_{\boldsymbol{\theta}}(\bar{y}|x) \nabla_{\boldsymbol{\theta}} \log p_{\boldsymbol{\theta}}(\bar{y}|x)^T) \tag{3}$$

The Fisher information matrix is used to select a small subset of important parameters $\hat{\boldsymbol{\theta}} \subseteq \boldsymbol{\theta}$ by sampling with probability distribution proportional to the Fisher information given by $p(\theta_i) = \frac{\mathcal{F}(\theta_i)}{\|\mathcal{F}(\boldsymbol{\theta})\|}$. We compute the gradient embeddings (Fisher score) of the unlabeled dataset with $g_u = \nabla_{\hat{\boldsymbol{\theta}}} \log p_{\boldsymbol{\theta}}(\bar{y}|x), x \in X_U$ and the query set with $g_q = \nabla_{\hat{\boldsymbol{\theta}}} \log p_{\boldsymbol{\theta}}(\bar{y}|x), x \in X_Q$. The similarity matrix is than constructed using Equation 4.

$$S_{uq} = g_u^T \mathcal{F}^{-1} g_q \tag{4}$$

## Appendix B.  USIMF-based gradient embedding visualization

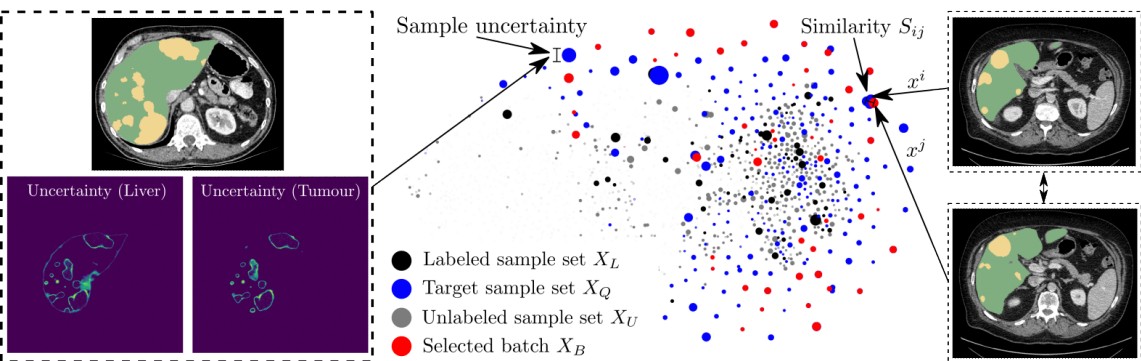

Figure 3: Visualization (t-SNE) of gradient embeddings from USIMF. Black dots: labeled dataset, grey dots: unlabeled dataset, blue dots: target dataset, red dots: selected batch samples. The diameter of the dots visualizes the sample uncertainty.

## Appendix C. Sampling strategies and implementation details

**Sampling strategies** We compare the following sampling method in our analysis: (1) Random Sampling, (2) Max Entropy (Shannon, 1948), (3) Mean STD (Kendall et al., 2017), (4) BALD (Gal et al., 2017), (5) Core-Set (Sener and Savarese, 2018), (6) BADGE(LL) (Ash et al., 2020; Aklilu and Yeung, 2022), (7) Stochastic Batches (Gaillochet et al., 2023) and (8) USIMC (ours), (9) USIMF (ours).

For Core-Set, the activation map of the last convolutional layer of the encode was used as feature vector to compute Euclidean distances between samples. BADGE(LL) is an adaptation of BADGE for segmentation tasks. The method uses the gradients from the last convolutional layer of the decoder to extract gradient embeddings. The method is equivalent to the ALGES-img method proposed in Aklilu and Yeung (2022). Stochastic Batches is a method that combines uncertainty and representation by measuring uncertainty at the level of batches instead of samples. The method depends on the number of batches $Q$ that are generated from the unlabeled dataset and we set $Q = floor(|D_U|/B)$ and used Entropy as uncertainty measure in our experiments. For a fair comparison between USIMC and USIMF, and to make the processing tractable, the gradient embeddings are truncated to a length of 10k elements.

**Implementation** The methods were evaluated on the first two folds of the nnUNet-based self-configured five-fold cross validation. The model was initialized with 100 randomly selected samples (patches) and the labeled dataset was expanded with a budget of $B = 100$ samples in each sampling round. In our experiments, we selected an annotation budget of 100 slices, since it can be considered a realistic budget for practical active learning scenarios. A larger budget may lead to less cost reduction, while a smaller budget may be impractical in realistic scenarios where radiologists must integrate the annotation process into their daily clinical routine. However, we evaluated our proposed strategy for larger budgets by increasing the budget to 900 and 2000 samples for the Hippocampus and Liver datasets, respectively, in the last sampling rounds even it might not be reasonable for practical active learning.

## Appendix D. Analysis of USIM based on query set selection strategy, batch uncertainty, query set size and weighting parameter

To further analyze and validate the proposed USIM sampling strategy, we investigated the query set selection strategy, batch uncertainty, influence of the query set size, and weighting parameter $\eta$. We only conducted experiments for the USIMF method since it can be considered the best performing method without loss of generalization.

**Query set selection:** We compared our proposed query set selection strategy based on the probability distribution proportional to the weighted uncertainty, with uniform sampling (similar to standard active learning in SIMILAR) (Kothawade et al., 2021) and maximum uncertainty-based sampling. We compared query set selection strategies for the first two sampling rounds on the Hippocampus and Liver datasets, as shown in Figure 4. We observed that our query set selection strategy (USIMF), outperformed uniform sampling $\boldsymbol{X_Q} \sim Uniform(\boldsymbol{X_U}, Q)$ (USIMF uniform sampling) and sampling based on maximum uncertainty $\boldsymbol{X_Q} \leftarrow \mathrm{argmax}_{\boldsymbol{X_Q} \subseteq \boldsymbol{X_U}, |\boldsymbol{X_Q}|=Q} \sum_{x_i \in \boldsymbol{X_Q}} Uncert(x_i)$ (USIMF max uncertainty sampling). Similar observations were made in (Gaillochet et al., 2023), where uncertainty-based sampling was performed on a stochastic batch level to overcome the shortcomings of pure uncertainty-based methods in sampling redundant data.

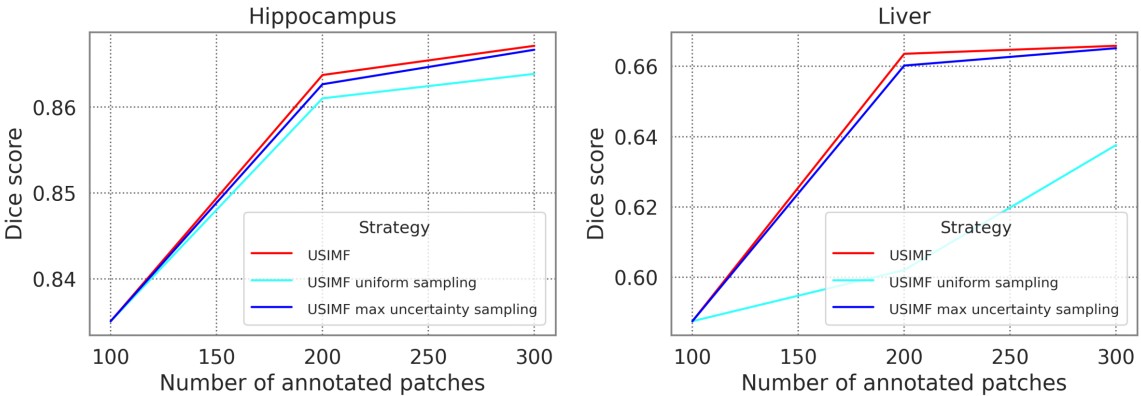

Figure 4: Performance gain in terms of dice score for our query set selection strategy (USIMF), query set selection using uniform sampling from the unlabeled dataset (USIMF uniform sampling) and sampling based on maximum uncertainty (USIMF max uncertainty sampling) on the Hippocampus and Liver dataset.

**Uncertainty:** We analyzed the amount of uncertainty of the selected batches to confirm that the chosen samples are indeed uncertain. We compared in Figure 5 the overall uncertainty $Uncert(\boldsymbol{X_B}) = \sum_{x \in \boldsymbol{X_B}} \sum_c \cdot Uncert(\boldsymbol{\theta}_{k-1}, \boldsymbol{x}, c)$ of the selected batches estimated using Monte Carlo dropout with $c$ being the foreground class and $\theta_{k-1}$ the trained model from the previous training round. The analysis was performed in the first five sampling rounds of the Hippocampus dataset. We observed that the uncertainty of batches sampled with USIMF and USIMC are lower than uncertainty based methods (Mean STD, ENTROPY, BALD) but higher than representativity-based methods such as CORESET, hybrid variants (BADGELL, STB) or Random sampling.

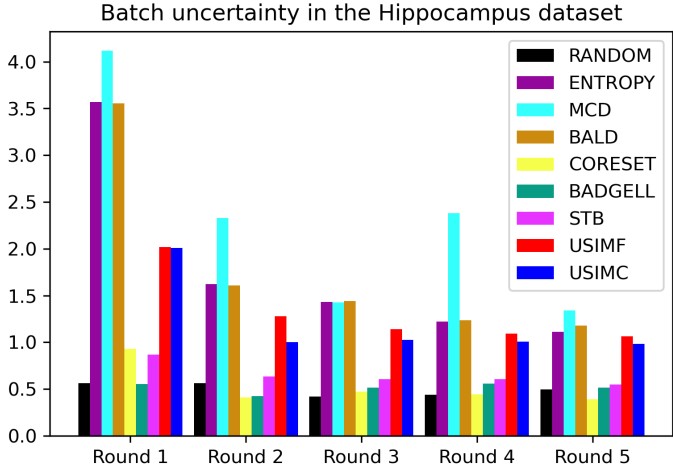

Figure 5: Overall batch uncertainty of the AL methods in the first five sampling rounds on the Hippocampus dataset.

To investigate the relationship between the segmentation error and the uncertainty measured by the gradient length and prediction uncertainty, we also performed a correlation analysis during the first sampling round of the Hippocampus dataset. We analyzed the segmentation error (total number of misclassified pixels) and predictive uncertainty, as well as the gradient length (L2 norms) of the bottleneck layer on a randomly selected subset of the unlabeled dataset. The correlation coefficients shows that for segmentation tasks with the nnUNet, the predictive uncertainty estimated using the Monte Carlo dropout correlates more strongly with the segmentation error ($r = 0.84$) than with the gradient length ($r = 0.68$), indicating that predictive uncertainty is a better measure of uncertainty.

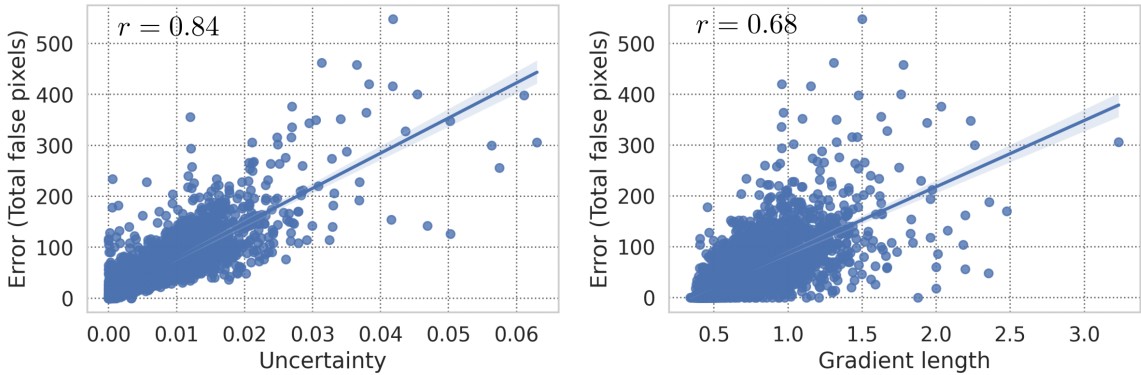

Figure 6: Correlation between the total number of misclassified pixels and the predictive uncertainty (left), and the correlation between the total number of misclassified pixels and the gradient length of the bottleneck layer (right)

**Query set size:** To analyze the influence of the size of the query set $Q$, we compare the performance gain with respect to $Q$ in Figure 7 for USIMF. For the Hippocampus dataset, we set $Q \in \{200, 400, 1600, U\}$ with $U$ being the size of the unlabeled dataset and compare it with automated estimation of the query set size (USIMF). The query set size for the USIMF method which was estimated with the elbow method (Thorndike, 1953) was $Q = 942$ (942/7662) and $Q = 1470$ (1470/7562) for the first and second round, respectively. For the Liver dataset, we set the query set size $Q \in \{200, 400, 800, 1600, 10000\}$. For the USIMF method, the estimated query size was automatically set to $Q = 6606$ (6606/104133) and $Q = 6852$ (6852/104033) for the first and second rounds, respectively. As the query set is sampled with replacement, we assume that the overall uncertainty of the batch should remain similar. It can be observed that the performance gain is not significantly affected by the size of the query set.

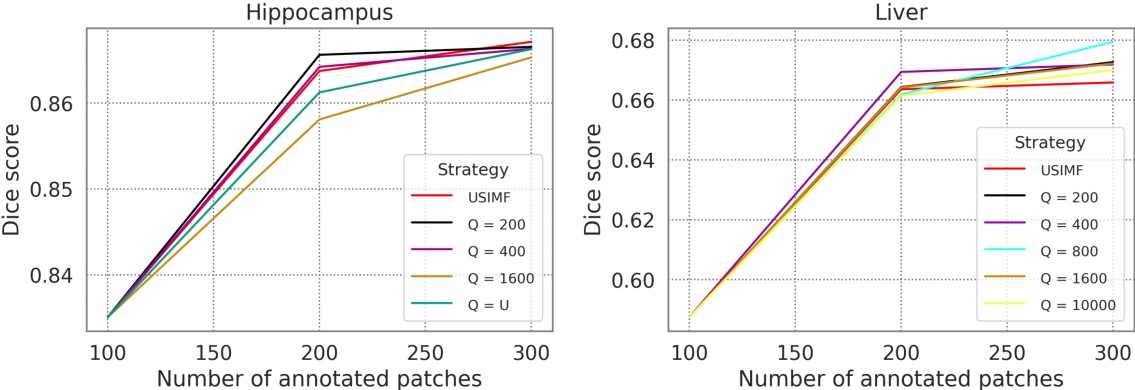

Figure 7: Performance gain with respect to the size of the query set $Q$ in the first two sampling round for the Hippocampus and Liver dataset.

**Weighting parameter $\eta$:** We analyze the influence of weighting parameter $\eta$ governing trade-off between query-relevance and diversity for USIMF. Kothawade et al. (2022b) have shown in their experiments that larger weighting parameters $\eta$ tend to increases query relevance while decreasing query coverage and diversity. We anlyzed the performance gain with respect to $\eta \in \{0.0, 1.0, 10.0, 100.0\}$ in the first two sampling rounds and did not encounter larger performance differences.

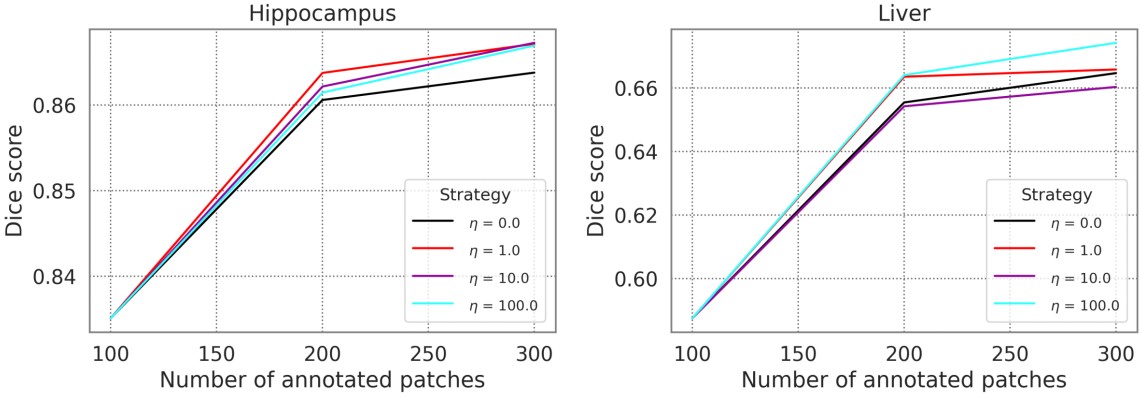

Figure 8: Performance gain with respect to the weighting parameter $\eta$.

## Appendix E. Redundancy scenario

To analyze our proposed method under a realistic redundancy scenario, we create a custom unlabeled Hippocampus dataset by dublicating all samples (patches) of one MRI scan (hippocampus_172.nii.gz) 100× after the initial training round. After the dublication process, the query set with a query size of $Q = 1000$ is sampled with probability distribution proportional to uncertainty. After sampling process, the query set consists of 45% samples from the same MRI scan. After submodular subset selection the selected batch $X_B$ includes only 18%(18/100) samples from the dublicated scan compared to 45%(45/100) by uniform sampling from the query set. It confirms that the proposed strategy with mutual information based submodular subset selection is robust with respect to redundancy. We compare the performance gain in terms of dice score based on the redundant unlabeled dataset for USIMF and uniform sampling from the query set in Figure 9.

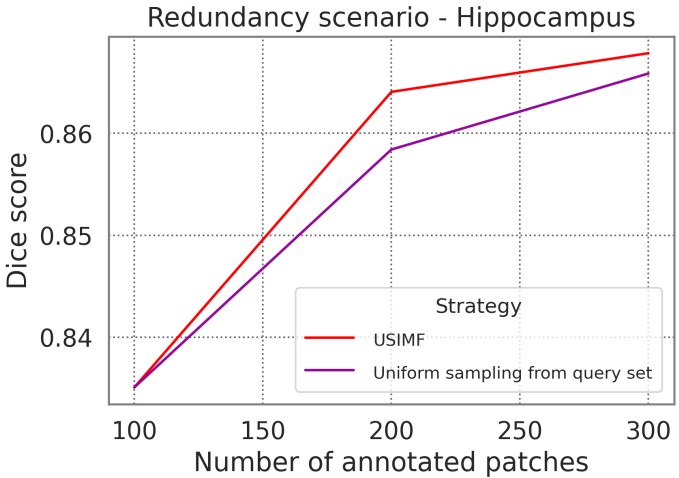

Figure 9: Performance comparison of USIMF and uniform sampling from the query set in a redundancy scenario.

## Appendix F. Configurations and self-configured hyperparameters by the nnUNet pipeline

Table 1: Configurations and self-configured hyperparameters

| Spleen dataset | |
|---|---|
| Hardware | 120GB RAM; NVIDIA A100-PCIE GPU, 40GB |
| nnUNet configuration | 2D |
| # downsampling stages | 8 |
| # model parameters | 10.2M |
| Batch size | 12 |
| Patch size | 512 x 512 |
| Epochs per training round | 300 |
| Loss function | Compound loss (dice and cross entropy loss) |
| Liver dataset | |
| Hardware | 120GB RAM; NVIDIA A100-PCIE GPU, 40GB |
| nnUNet configuration | 2D |
| # downsampling stages | 8 |
| # model parameters | 46.3M |
| Batch size | 12 |
| Patch size | 512 x 512 |
| Epochs per training round | 300 |
| Loss function | Compound loss (dice and cross entropy loss) |
| Hippocampus dataset | |
| Hardware | 120GB RAM; NVIDIA V100 GPU, 32GB |
| nnUNet configuration | 2D |
| # downsampling stages | 4 |
| # model parameters | 1.9M |
| Batch size | 366 |
| Patch size | 56 x 40 |
| Epochs per training round | 300 |
| Loss function | Compound loss (dice and cross entropy loss) |

## Appendix G. Examples of active learning for medical image segmentation

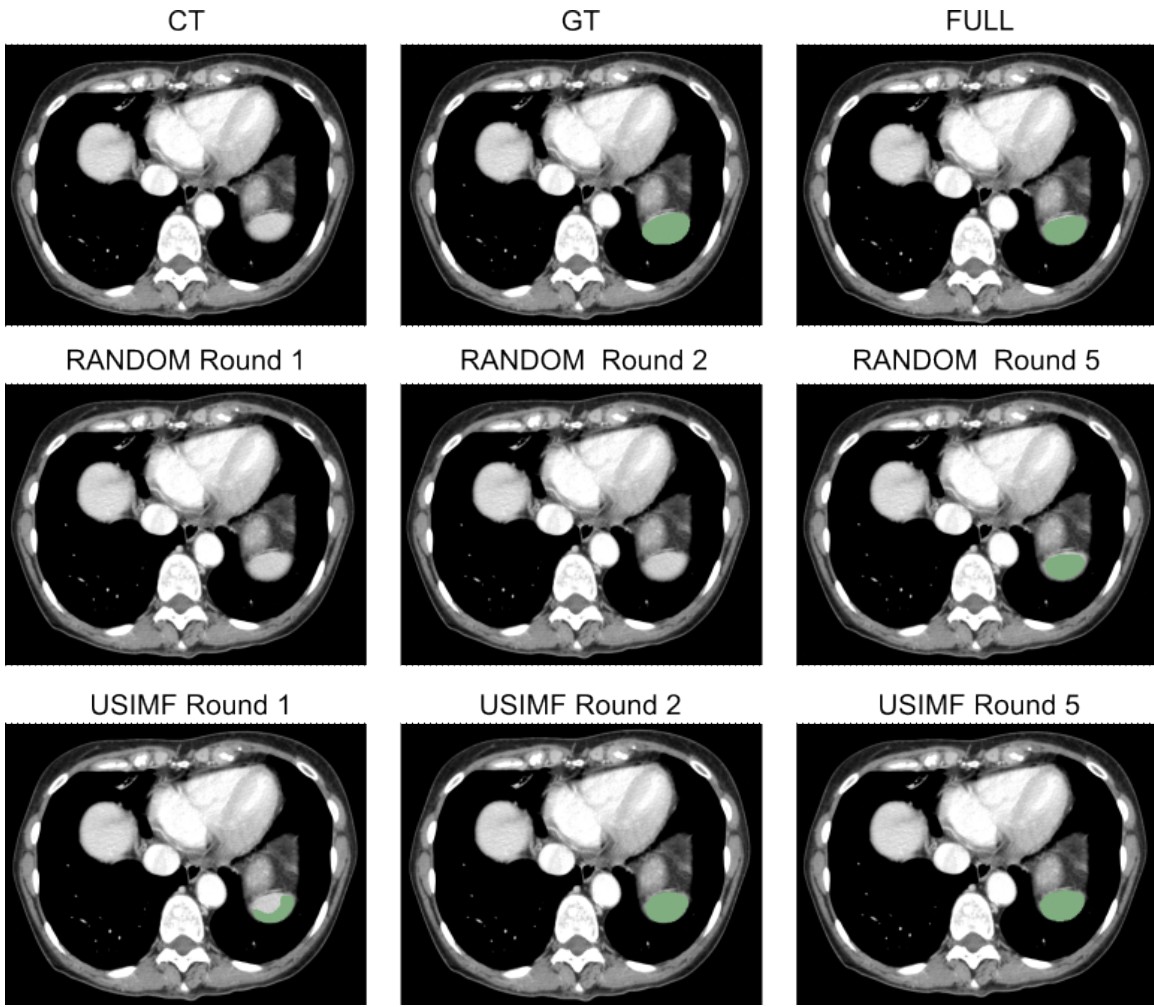

Figure 10: The top row, from left to right, shows a CT slice of the spleen, the ground truth segmentation, and the segmentation result of the model train on the fully annotated dataset. The second row shows the segmentation results after the first and fifth round of random sampling. The third row shows the segmentation results after the first and fifth round of the USIMF sampling method.

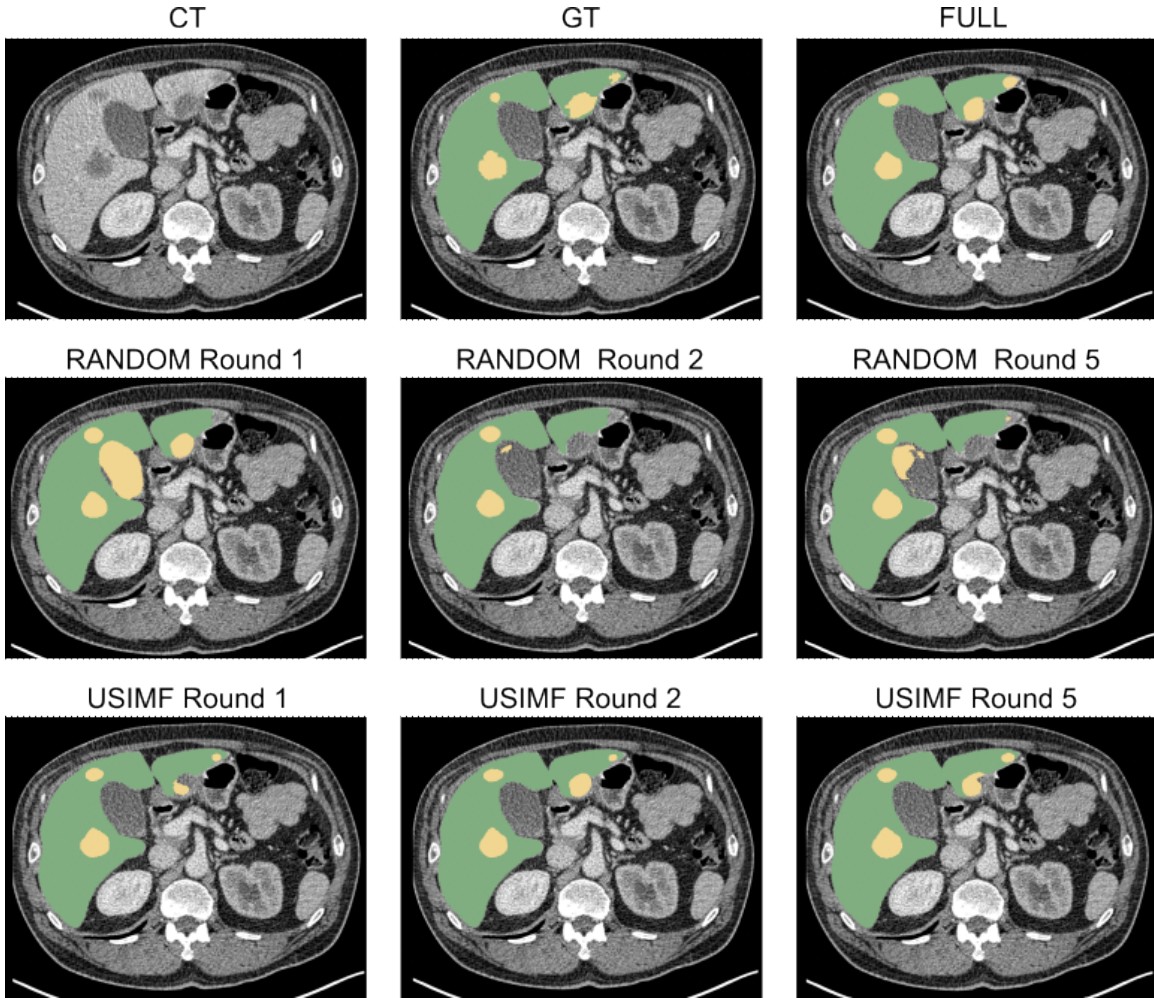

Figure 11: The top row, from left to right, shows a CT slice of the liver, the ground truth segmentation, and the segmentation result of the model train on the fully annotated dataset. The second row shows the segmentation results after the first and fifth round of random sampling. The third row shows the segmentation results after the first and fifth round of the USIMF sampling method.

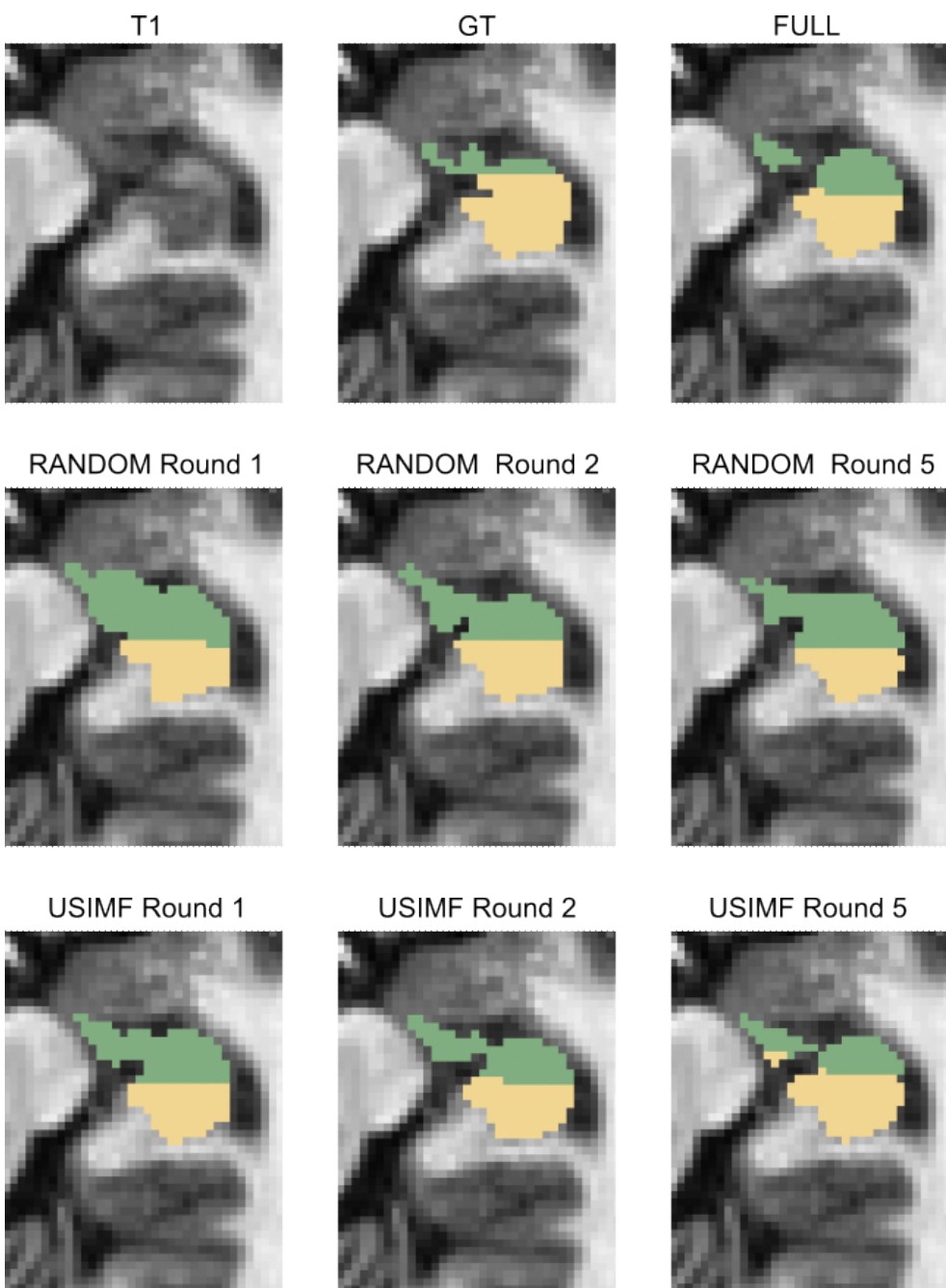

Figure 12: The top row, from left to right, shows an MRI slice of the hippocampus, the ground truth segmentation, and the segmentation result of the model train on the fully annotated dataset. The second row shows the segmentation results after the first and fifth round of random sampling. The third row shows the segmentation results after the first and fifth round of the USIMF sampling method.

