# OpenReview forum: "Active Learning with the nnUNet and Sample Selection with Uncertainty-Aware Submodular Mutual Information Measure"
_MIDL.io/2024/Conference — MIDL 2024 Poster_

### Official Review · Reviewer_9u9e · 2024-02-23

**Confidence:** 4
**Preliminary Rating:** 4
**Recommendation:** Oral

**Summary:**

This paper introduces USIM (Uncertainty-Aware Submodular Mutual Information Measure), a novel sample selection strategy tailored for UNet-like models. USIM is designed to select batches of uncertain, diverse, and representative samples under active learning settings. The experiments confirms that the USIM approach outperforms traditional methods in three distinct medical image segmentation tasks.

**Strengths:**

This study presents USIM (Uncertainty-Aware Submodular Mutual Information Measure), an innovative approach for selecting samples specifically designed for UNet-like architectures. USIM aims to identify groups of samples that are uncertain, diverse, and representative within the framework of active learning. Experiments show that the USIM method surpasses conventional techniques across three separate tasks of medical image segmentation.

**Weaknesses:**

I understand the general concepts of the methods, but I struggle with some specifics of the model. The authors argue that their method excels by selecting batches of samples that are uncertain, diverse, and representative. Yet, I find a lack of both quantitative and qualitative evidence to demonstrate that the samples indeed meet these criteria.

**Detailed Comments:**

1. In section 3.1.2, the authors claim that "maximizing the SMI function ensures the sampling of diverse and relevant samples with respect to the query set." I am unable to understand why this is the case. If the SMI function measures the mutual information, it looks like maximizing SMI will identify the set $X_B$ that are the most similar to $X_Q$, but it is unclear why $X_B$ is diverse.

2. The authors seem to claim the Equation (1) combines uncertainty and submodular subset selection. However, it is unclear why this equation is submodular. Does it depend on the form of the similarity function? It is also unclear why this function measures uncertainty.

3.  The authors argue that their method excels by selecting batches of samples that are uncertain, diverse, and representative. Yet, I find a lack of both quantitative and qualitative evidence to demonstrate that the samples indeed meet these criteria.

**Justification Of The Preliminary Rating:**

The USIM method introduced in this paper appears to be innovative. Experiments demonstrate that USIM exceeds the performance of traditional methods in three separate medical image segmentation tasks. I believe that providing a clearer explanation of the problem mentioned earlier would enhance the overall presentation.

**Questions To Address In The Rebuttal:**

1. Why does maximizing the SMI function ensuresthe sampling of diverse and relevant samples with respect to the query set?

2. Why is Equation 1 submodular? Why does this function measure uncertainty?

3. Are there quantitative and qualitative evidence to demonstrate that the samples selected are uncertain, diverse, and representative?

**Special Issue:**

No

---

> ### Author Response · Authors · 2024-03-17
> **Thank you very much for your constructive comments.**
>
> **Weaknesses:**
>
> > I understand the general concepts of the methods, but..
>
> In order to confirm our hypothesis that our selected samples are indeed uncertain, diverse, and representative, we conducted additional experiments in Appendix D.
> To verify that the samples are uncertain, we analyzed the uncertainty estimated using Monte Carlo dropout. We compared the overall uncertainty of the selected batches between all sampling strategies in Appendix D, Uncertainty.
> One big advantage of our subset selection strategy is the robustness with respect to sample redundancy in the unlabeled dataset. To confirm that the samples are diverse and representative, we performed an experiment to analyze their behavior under strong sample redundancy in the unlabeled dataset in Appendix D, Redundancy.
> As described in Kothawade et al., (2022), the FLQMI function can balance query-relevance on one hand and query-coverage and diversity on the other hand by adjusting the weighting parameter $\eta$. The influence of $\eta$ with respect to the performance gain was studied in Appendix D, Weighting parameter $\eta$, and we have clarified this in the text and updated equation (1).
>
> > 1. In section 3.1.2, the authors claim...
>
> To confirm that the selected samples are indeed diverse, we study the performance gain under strong sample redundancy in the unlabeled dataset in Appendix D, Redundancy. It shows that the SMI function selects less redundant samples than uniform sampling from the query set, which is a major advantage of our method.
> The FLQMI function has the ability to balance query relevance on the one hand and query coverage and diversity on the other hand by changing a weighting parameter $\eta$. We have clarified this in the text and updated Eq. (1) with the weighting parameter $\eta$. We also study the influence of the weighting parameter in Appendix D. We refer the interested reader to Beck et al. (2024) for a more theoretical analysis of the submodular mutual information measure and the FLQMI function. Diversity is further supported by sampling the query set with a probability distribution proportional to the sample uncertainty instead of sampling the most uncertain samples.
>
> > 2. The authors seem to claim the Equation (1) ...
>
> We apologize for the confusion in the text, and we restructured the section to present the general idea of submodular mutual information measure for subset selection in section 3.1.2 and highlight our adaptations and contributions in section 3.3.
> The submodular mutual information (SMI) function is given by
> $I_f(X_B ; X_Q)=f(X_B)+f(X_Q)-f(X_B \cup X_Q)$, where $X_Q$ is the query set $X_B$ is the selected subset and $f: 2^{X} \rightarrow \mathbb{R}$ a submodular set function defined as
> $f(X)=\sum_{x_i \in \Omega}\max_{x_j \in X} S_{i j}$.
> The Equation (1) is a mutual information function variant (FLQMI) for the facility location (FL) function defined over the query set $X_Q$. Intuitively, the function measures the similarity between the query set $X_Q$ and the selected subset $X_B$.
> The function does not measure uncertainty. However, the query set consists of samples with high predictive uncertainty. These samples are sampled from an unlabeled dataset with a probability distribution proportional to predictive uncertainty. Therefore, we assume that samples selected in $X_B$ should also have high uncertainty. We confirmed this hypothesis by comparing all sampling strategies with respect to the overall uncertainty of selected batches in Appendix D. Kothawade et al. (2022) analyzed in detail several submodular information measures for guided subset selection.
>
> > 3. The authors argue that their method excels by selecting batches of samples that are uncertain, diverse, and representative. Yet, I find a lack of both quantitative and qualitative evidence to demonstrate that the samples indeed meet these criteria.
>
> We performed additional experiments to confirm that selected batches are uncertain, diverse, and representative. In Appendix D, we compare the batch uncertainty of the sampling methods. In addition, we analyzed the behavior under sample redundancy in the unlabeled dataset to demonstrate that the sampling methods selected samples that are diverse and representative.
> Moreover, as described in Kothawade et al., (2022), the FLQMI function has the ability to balance query relevance on the one hand and query coverage and diversity on the other hand by changing a weighting parameter $\eta$. We have clarified this in the text and updated Eq. (1) with the weighting parameter $\eta$. We study the influence of the weighting parameter in Appendix D.
>
> Questions To Address In The Rebuttal:
>
> > 1. Why does maximizing the SMI function...?
>
> See answer above.
>
> > 2. Why is Equation 1 ...
>
> See answer above.
>
> > 3. Are there quantitative and qualitative evidence...
>
> See answer above.

---

### Official Review · Reviewer_Biq8 · 2024-02-26

**Confidence:** 4
**Preliminary Rating:** 3
**Final Rating:** 4

**Summary:**

The paper presents a active learning (AL) approach for medical image segmentation based on the nnUNet model and sample selection with submodular measures of mutual information. This approach exploits the pre-processing and planning capabilities of nnUNet to select optimal hyperparameters for each segmentation task, in an automated way. The active learning strategy used in the approach follows the SIMILAR method (Kothawade et al., 2021), modifying this method in two ways: 1) applying it to segmentation instead of classification, 2) selecting the query set based on Monte-Carlo dropout uncertainty estimation. Experiments on three segmentation tasks/datasets (Spleen, Liver and Hippocampus) show the proposed method to outperform several recent AL algorithms.

**Strengths:**

* Results show improvements over compared methods in all nearly all test cases.

* The paper compares two variants of the method, based on cosine similarity of gradient embeddings (USIMC) and on the Fisher kernel (USIMF). Results demonstrate the better performance of USIMF.

* With the exception of a few typos, the paper is well written.

**Weaknesses:**

* One of the main weaknesses of the paper is its limited methodological novelty. The proposed method mostly adapts the SIMILAR algorithm to segmentation using nnUnet as base network. The uncertainty aware selection of the query set is interesting, however it can be seen as a variant of the one used in (Kothawade et al., 2021) for rare classes, and the usefulness of this strategy is not properly evaluated in the experiments.

* Given that one of the main differences is the selection of the query set, I expected a more thorough validation of this strategy. Currently, it is hard to tell if this strategy brings any benefit.

**Detailed Comments:**

* p2: thr method --> our method ?

* p2: "we propose and evaluate USIM, an AL strategy that combines predictive uncertainty with diversity and representativeness in the parameter space, using a submodular mutual information measure." Authors should mention explicitly the novel contributions w.r.t. SIMILAR

* What is the size of the query set, and does this size affect performance? What is the performance if using X_Q = X_U as in the default AL scenario of SIMILAR?

* Algorithm 1: Do you allow selection samples from X_Q. If so, isn't Eq (1) maximized by setting X_B ~= X_Q ?

* p6: encode was --> encoder was

**Justification Of Final Rating:**

The authors have added several ablation experiments showing the benefit of their query sample selection strategy, as well as the impact of query set size and hyper-parameter eta. Given these improvement, I upgrade my score to weak accept (however, not strong accept since I still believe that the proposed method is mainly an extension of SIMILAR to segmentation).

**Justification Of The Preliminary Rating:**

As justification, I refer to the main weaknesses mentioned above:

* One of the main weaknesses of the paper is its limited methodological novelty. The proposed method mostly adapts the SIMILAR algorithm to segmentation using nnUnet as base network. The uncertainty aware selection of the query set is interesting, however it can be seen as a variant of the one used in (Kothawade et al., 2021) for rare classes, and the usefulness of this strategy is not properly evaluated in the experiments.

* Given that one of the main differences is the selection of the query set, I expected a more thorough validation of this strategy. Currently, it is hard to tell if this strategy brings any benefit.

**Questions To Address In The Rebuttal:**

Clarify the novel contributions wrt SIMILAR and evaluated (if possible, else discuss) the impact of the query set selection strategy.

---

> ### Author Response · Authors · 2024-03-17
> **Thank you so much for your constructive comments, please see the detailed point-by-point response below.**
>
> **Weaknesses:**
> > One of the main weaknesses of the paper is its limited methodological novelty. The proposed method mostly adapts the SIMILAR algorithm to segmentation using nnUnet as base network. The uncertainty aware selection of the query set is interesting, however it can be seen as a variant of the one used in (Kothawade et al., 2021) for rare classes, and the usefulness of this strategy is not properly evaluated in the experiments.
>
> We appreciate the reviewer for bringing to our attention the need for clarification regarding the novelty of our method.
> Our contributions are twofold. Firstly, we select the query set based on the weighted predictive uncertainty using Monte Carlo dropout. To further evaluate this strategy, we performed additional experiments and compared our method with selecting the query set based on a uniform distribution over the unlabeled dataset (standard active learning). In addition, we compared our strategy to selecting the most uncertain samples from the unlabeled dataset as the query set. The results of the experiments are presented in Appendix D and show that our query set selection strategy is more efficient.
> Secondly, SIMILAR was developed and evaluated solely for classification tasks, using last layer gradients as feature embedding. However, for U-Net like architectures, the last layer gradients do not guarantee to provide an accurate representation for estimating similarities between samples.
> We analyzed two variants where the similarity matrix is estimated based on the cosine similarity of the gradients of the bottleneck layer, and a variant where the gradients of the decoder are selected based on the Fisher information measure.
> We further evaluate our method in a redundancy scenario in Appendix E to confirm the superiority of the sampling method compared to uniform selection from the query set..
>
> > Given that one of the main differences is the selection of the query set, I expected a more thorough validation of this strategy. Currently, it is hard to tell if this strategy brings any benefit.
>
> We agree that the selection of the query set is novel. The query set was chosen by sampling from a probability distribution over the unlabeled dataset, proportionally to the weighted predictive uncertainty. In Appendix D, Query set selection, we conducted a more comprehensive validation by selecting samples with a uniform probability distribution over the unlabeled dataset (standard active learning) and choosing the most uncertain samples.
>
> > p2: thr method --> our method ?
>
> Thank you, we corrected the typo.
>
> **Detailed Comments:**
>
> > p2: "we propose and evaluate USIM, an AL strategy that combines predictive uncertainty with diversity and representativeness in the parameter space, using a submodular mutual information measure." Authors should mention explicitly the novel contributions w.r.t. SIMILAR
>
> Thank you for your comment. We have clarified the novelty of our method in the introduction, as described above.
>
> > What is the size of the query set, and does this size affect performance? What is the performance if using X_Q = X_U as in the default AL scenario of SIMILAR?
>
> Thank you for your comment. In Appendix D, Query set size, we performed additional experiments to analyze the performance gain of our method with respect to the size of the query sets and could show that the query size does not affect the performance by a large margin. We also can assume that the overall uncertainty of the batch should remain similar since we select samples from the unlabeled dataset with replacement. In our experiments we estimated the size of the query set with the simple elbow method.
>
> > Algorithm 1: Do you allow selection samples from X_Q. If so, isn't Eq (1) maximized by setting X_B ~= X_Q ?
>
> Yes, we allow selection of samples from $X_Q$ and we changed the text to make this clear. In our experiments, the size of the query set $X_Q$ was always larger than the sampling budget. We made this clear in the text.
>
> > p6: encode was --> encoder was
>
> Thank you, we corrected the typo.
>
> **Questions To Address In The Rebuttal:**
>
> > Clarify the novel contributions wrt SIMILAR and evaluated (if possible, else discuss) the impact of the query set selection strategy.
>
> Thank you for your comment. We have clarified the novel aspects and contributions in the introduction and evaluated the influence of query set selection strategy through various experiments in Appendix D. These experiments analyzed the influence of the query set selection strategy and compare it with selecting the query set based on a uniform distribution over the unlabeled dataset (standard active learning) and selection of most uncertain samples. We further analyze batch uncertainty over sampling strategies, influence of the query set size and influence of the weighting parameter $\eta$.
> We additionally analyze exemplary the robustness with respect to redundancies in the unlabeled dataset in Appendix E.

---

### Official Review · Reviewer_PcXq · 2024-03-11

**Confidence:** 5
**Preliminary Rating:** 3
**Recommendation:** Poster
**Final Rating:** 4

**Summary:**

The paper addresses the high annotation cost in image segmentation tasks through an active learning strategy. To this end, the paper proposes to
- evaluate the feasibility of a self-configuring nn-UNet  network.
- guide the active learning sampling strategy with a measure combining a monte-carlo uncertainty from (Gal 2016, 2017) and  the diversity and representative measures from  (Kothawade et al., 2021, 2022). The later is computed on parameter space with a submodular mutual information SMI measure.

**Strengths:**

- The annotation cost problem for segmentation is real
- The code has been made available
- The evaluation is done on three organs of the public decathlon dataset (spleen, liver, hippocampus)
- Gradient embeddings have been used before to measure model representativennes but mostly for classification tasks (Kothawade et al., 2021, 2022), here the paper proposes to use them  for segmentation
- The paper evaluates two variants of the SMI (one based on a the last CNN layer and a second based on Fisher Information)

**Weaknesses:**

- The state of the art and evaluation do not discuss pseudolabeling strategies, which have been largely used in the field to tackle the cost of annotations.
- Appart of using gradient embeddings for segmentation, are there other novel aspects?
- The evaluation is done only with 2D models and single organ tasks.
- The arguments behind individual method choices are not clear and not always in phase with the introduction motivations.
- The proposed pipeline proposes to reconfigure the network each round of sampling which could raise a convergence issue.
- The practicality of Active Learning remains to be proven in real scenarios, specially how to determine a stopping criteria when GT is really missing?

**Detailed Comments:**

- The state of the art and evaluation do not discuss pseudolabeling strategies, which have been largely used in the field to tackle the cost of annotations.

- Appart of using gradient embeddings for segmentation, are there other novel aspects?

- The evaluation is done only with 2D models for 3D segmentation tasks. In particular, I disagree with the argument by the authors that one should opt for 2D segmentation models given “the lack of guaranteed performance improvements of 3D models” . It is certainly more efficiently to perform experiments with 2D models, but this choice goes against most recent organ segmentation methods which in addition also target the more challenging multi-organ task.

- The introduction and related work motivate the approach by focusing on the self-configuring properties of the nn-UNet to justify the reproducibility, robustness, and accessibility to non-experts. However, it is not clear what the method is robustness against exactly? also, the method requires complex steps which may not be deployed by non experts (e.g. monte-carlo dropout, similarity matrices for the SMI measure, or the implementation selection approaches).  These steps also contrast with the claim that the proposed method is simpler than than approaches based on changing the learning loss (Yoo and Keweon 2019)
- “Sampling, annotation and self-configured model finetuning is repeated in multiple AL round” . Could reconfiguration raise an convergence issue?
- The number of initial samples seems realistic but going up to 900 or to 2000 as it is done for some experiments seems unreasonable for an AL scenario
- The practicality of Active Learning remains to be proven in real scenarios, if no GT is available how to determine a stopping criteria?

**Justification Of Final Rating:**

Several aspects of the paper have been clarified, in particular the novelty. Although there are some limitations regarding the scope of the evaluation, new results presented after the rebuttal should ensure interesting discussions at MIDL

**Justification Of The Preliminary Rating:**

The problem is interesting, the approach follows recent trends, but the evaluation set up is limited to a 2D and single organ case. There are also some clarity and motivation issues, as well as questions to be addressed in the rebuttal.

**Questions To Address In The Rebuttal:**

Comments
- The number of initial samples seems realistic but going up to 900 or to 2000 as it is done for some experiments seems unreasonable for an AL scenario
- Gradient embeddings is mentioned to be promising as it works for classification, it was not discussed why is it interesting in the case of segmentation
- The stochastic greedy method in step 7 of the algorithm was not explained
- clearly state in the related work which methods have been used for segmentation and which for classification only.



Questions
- The method extracts overlapping patches. Why is this the case? can there be an overlap between a labelled and an unlabelled example?
- Why relying on Monte-Carlo dropout to compute the uncertainty which requires several evaluations of the method. Why not using the softmax predictions to compute the uncertainties?
- Why are the results in Fig 2 not complete for other methods? Only the curves of SMI methods go beyond a number of samples
- How do evaluated methods compare in terms of computational complexity and memory?

Typos
“samples, thEIr method is using gradient embeddings”
“training of additional models to evaluate informativeness of unlabeled sampleS, such”

**Special Issue:**

No

---

> ### Author Response · Authors · 2024-03-17
> **Thank you very much for your detailed and constructive feedback.**
>
> **Weaknesses**:
>
> > The state of the art and evaluation do not discuss pseudolabeling strategies...
>
> We have revised the state-of-the-art section and added these strategies.
>
> > Appart of using gradient embeddings...
>
> We revised the manuscript and clarified the novelty of our method.
> Our novel contributions are twofold.
> Firstly, we select the query set based on the weighted predictive uncertainty using Monte Carlo dropout which can be seen as a variant of SIMILAR for rare classes. To evaluate this strategy, we compared it with selecting the query set based on a uniform distribution over the unlabeled dataset (standard active learning). In addition, we compared our strategy to selecting the most uncertain samples from the unlabeled dataset as the query set. The results of the experiments are presented in Appendix D and show that our query set selection strategy is more efficient.
> Secondly, SIMILAR was developed and evaluated solely for classification tasks, using last layer gradients as feature embedding. However, for U-Net like architectures, the last layer gradients do not guarantee to provide an accurate representation for estimating similarities between samples. We propose two variants where the similarity matrix is estimated based on the cosine similarity of the gradients of the bottleneck layer, and a variant where the gradients of the decoder are selected based on the Fisher information measure.
>
> > The arguments behind individual method choices..
>
> We added more experiments to analyse and explain individual method choices in the text.
>
> > The practicality of Active Learning remains...
>
> We agree that practicality of AL must be demonstrated in real scenarios in future reseacrch. Determining a stopping criteria remains an open question. We included the issue in the discussion.
>
> **Detailed Comments:**
>
> >  The evaluation is done only with 2D models ...
>
> For our experiments, we used 2D nnUNet configurations because of higher training speed and less prone to overfit. However, the method can be extended to 3D models and will be studied in further research. We added this as a limitation and rephrased the text.
>
> > The introduction and related work...
>
> We apologize for the misunderstanding. The motivation of this work is to evaluate AL methods in combination with the nnUNet model. The nnUNet is used since it identified robust design decisions based on multiple tasks. We reformulated the introduction and state-of-the art section.
> > “Sampling, annotation and self-configured...
>
> Since only the annotations change during each sampling round, almost all model configurations except annotation resampling and intensity normalization remain the same. We clarified that. We did not encounter any convergence issues in our experiments.
>
> **Questions To Address In The Rebuttal:**
> > The number of initial samples...
>
> The initial sample size of 100 is indeed realistic. However, we wanted to test the proposed strategy also for larger budgets to confirm its robustness with respect to the budget. We have clarified this in the text.
>
> > Gradient embeddings is mentioned...
>
> Gradient embeddings are a promising choice in AL for classification tasks by measuring how much a new data point influences changes in model parameters. Aklilu et al. (2022) adapted this idea for semantic segmentation tasks. However, the efficient extraction of gradient embeddings in U-Net like architectures remains an open question that we aim to address. We added this to the introduction section.
>
> > The stochastic greedy method...
>
> We utilized the stochastic greedy optimizer from the SUBMODLIB library developed to maximize the submodular function. We have clarified this in the text.
>
> > clearly state in the related work...
>
> We revised the related work section accordingly.
>
> **Questions**
>
> > The method extracts overlapping patches...
>
> Overlaps between labeled and unlabeled patches exist.
> However, the nnUNet provides an “ignore” label to ensure that only annotated regions (patches) are used to train the model.
>
> > Why relying on Monte-Carlo dropout...
>
> It has been shown that uncertainty estimation using Monte Carlo Dropout seem to outperform uncertainty estimation using softmax predictions for AL. Our experiments confirmed that findings. (see Fig. 2)
>
> > Why are the results in Fig 2...
>
> We only wanted to confirm that our proposed methods work well with larger budgets, and therefore did not show performance results for the other methods. We completed the curves for the other methods.
>
> > How do evaluated methods...
>
> Computational complexity and memory requirements were not analysed in detail. However, to ensure a fair comparison between USIMC and USIMF and to make the processing manageable, the gradient embeddings were truncated to a length of 10k elements.
> The complexity of submodular subset selection depends on kernel computation time O(Q\*U) and the complexity of the greedy algorithm O(Q\*U\*B). For more information we refer to Kothawade et al. (2021).

---

### Meta-Review · Area_Chair_N7zh · 2024-04-04

**Recommendation:** Accept (Poster)
**Confidence:** 4

**Metareview:**

The paper introduces an active learning (AL) strategy for medical image segmentation using the nnUNet model, focusing on reducing annotation costs. The approach incorporates submodular measures of mutual information for sample selection and leverages the pre-processing capabilities of nnUNet to optimize hyperparameters for each segmentation task automatically. While the approach shows promise in improving segmentation performance, reviewers raise concerns about methodological novelty, evaluation, and clarity in certain aspects.

While the paper presents a promising approach for reducing annotation costs in medical image segmentation, addressing the weaknesses highlighted by reviewers, such as conducting a more thorough evaluation, clarifying methodological choices, and comparing with existing approaches, would strengthen its contribution and impact in the field.

---

### Decision · Program_Chairs · 2024-04-06

Accept (Poster)